# Solve High-Dimensional Reflected Partial Differential Equations by Neural Network Method

**Xiaowen Shi [1], Xiangyu Zhang [2], Renwu Tang [1] and Juan Yang [3,\*]**

[1] School of Government, Beijing Normal University, Beijing 100875, China; wsygjdgm@163.com (X.S.); rwtang@bnu.edu.cn (R.T.)
[2] School of Information Science and Engineering, Shandong Agriculture and Engineering University, Jinan 251100, China; z2021080@sdaeu.edu.cn
[3] School of Science, Beijing University of Posts and Communications, Beijing 100876, China
\* Correspondence: 202231260019@mail.bnu.edu.cn

**Abstract:** Reflected partial differential equations (PDEs) have important applications in financial mathematics, stochastic control, physics, and engineering. This paper aims to present a numerical method for solving high-dimensional reflected PDEs. In fact, overcoming the "dimensional curse" and approximating the reflection term are challenges. Some numerical algorithms based on neural networks developed recently fail in solving high-dimensional reflected PDEs. To solve these problems, firstly, the reflected PDEs are transformed into reflected backward stochastic differential equations (BSDEs) using the reflected Feyman–Kac formula. Secondly, the reflection term of the reflected BSDEs is approximated using the penalization method. Next, the BSDEs are discretized using a strategy that combines Euler and Crank–Nicolson schemes. Finally, a deep neural network model is employed to simulate the solution of the BSDEs. The effectiveness of the proposed method is tested by two numerical experiments, and the model shows high stability and accuracy in solving reflected PDEs of up to 100 dimensions.

**Keywords:** neural network; reflected PDEs; high-dimensional problem; penalization method

## 1. Introduction

As a significant mathematical tool for characterizing singularities, the reflection theory and reflected PDEs have a broad range of applications in various fields, including physics, engineering, and finance. These applications encompass describing materials with memory and heredity, determining system equilibrium points, and pricing financial assets. Solving reflected PDEs can be considered as a Skorohod problem [1]. The reflection term in these equations restricts the solution above an obstacle and represents the minimum external force that prevents the solution from breaching the obstacle [2,3]. Since it is challenging to find an explicit solution to reflected PDEs, the development of efficient numerical algorithms is a crucial area of research.

There are many numerical methods for solving PDEs without reflection. For low-dimensional PDEs, practical algorithms include the neural network-based method [4], finite element method [5,6], finite difference method [7–9], Monte Carlo approximation method [10,11], spectral Galerkin approximation method [12,13], and sparse grid approximation method [14,15]. These methods have been proven effective and stable in a large body of literature. However, when it comes to high-dimensional problems, these traditional methods are often not applicable due to the "dimensional curse" [16], which causes an exponential increase in computational complexity and rapid reduction in stability and efficiency as dimension increases. Although the Monte Carlo method can overcome the "dimensional curse", it can only approximate the solution of an isolated point in space. Recently, with the development of deep learning technology, more and more deep learning-based numerical methods have been developed for solving PDEs, successfully overcoming the "dimensional

curse". For linear high-dimensional PDEs, Beck et al. transformed the problem into a stochastic control problem and designed a deep neural network-based calculator [17], while Becker et al. used a multi-layer Picard iteration method [18]. For high-dimensional nonlinear PDEs, advanced methods include the deep Galerkin method, deep splitting approximation method, nonlinear Picard iteration method and deep BSDE method [19–24]. Artificial neural network-based algorithms have demonstrated impressive computational capabilities in solving complex problems, including multi-parameter low-dimensional problems and high-dimensional problems [25,26]. Despite the significant progress made in solving high-dimensional PDEs, there remains a gap in research on the numerical solution of high-dimensional reflected PDEs. To address reflected problems, Cagnetti et al. utilized the adjoint method to examine the Hamilton–Jacobi equation obstacle problem [3], and Côme et al. applied deep learning algorithms to solve nonlinear parabolic reflected PDEs [4]. However, only the RDBDP algorithm based on deep neural networks proposed by Côme et al. [4] can currently solve high-dimensional reflected PDEs with dimensions up to 40. Other methods that have proven successful in solving high-dimensional PDEs are not capable of handling corresponding reflected problems.

The aim of this study is to present a novel numerical algorithm, known as the Deep C-N method, for solving high-dimensional parabolic reflected PDEs. Through the conversion of the problem into reflected BSDEs and penalization method, the reflection term is effectively incorporated into the numerical solution. The use of deep neural networks in the Deep C-N method provides a flexible and powerful approach to tackle complex and high-dimensional problems. This work builds upon previous research and expands upon the existing methods for solving reflected PDEs. This method is an improvement over the traditional deep neural network methods, which are used to solve PDEs and reflected PDEs. The key contribution of this work lies in the following aspects: 1. By utilizing the penalization method, the reflection term is approximated, rather than being directly simulated, as in Côme et al. [4]. 2. In comparison to the Deep BSDE method proposed by E et al. [22], Beck et al. [23], and Han et al. [24], the proposed Deep C-N method demonstrates a higher approximation accuracy in solving high-dimensional PDEs without reflection. This is evident from the numerical results obtained from the Allen–Cahn equation, which was tested up to 400 dimensions. 3. The experiment of pricing American options confirms that the Deep C-N method has the capability of solving reflection problems in higher dimensions compared to the RDBDP method proposed by Come et al. [4]. The test was conducted up to 100 dimensions.

The contents of this paper are structured as follows. Section 2 presents the formulation of the reflected PDEs, followed by a demonstration of the conversion of the reflected PDEs into BSDEs. Section 3 provides a comprehensive overview of the Deep C-N method, including its operational procedures and numerical experiments, which demonstrate its significant properties through the examination of high-dimensional Allen–Cahn and American option equations. Finally, Section 4 concludes the paper by summarizing the key findings and outlining potential avenues for future research.

## 2. Approximating Schemes for Reflected PDEs

In this section, we propose a numerical algorithm named the Deep C-N method for solving the obstacle problems for high-dimensional non-linear parabolic PDEs. Generally, an obstacle problem for PDEs can be totally solved via three steps based on the above method. In the first step, a connection will be built between as the solution for obstacle problems for PDEs and that of the corresponding reflected BSDEs by the non-linear reflected Feynman–Kac formula. As a result, the original problems for PDEs are converted to problems for RBSDEs. In the second step, the problems for RBSDEs will be transformed further. In this step, the issue about RBSDEs will be considered as an optimal stopping-time problem via the penalty approach. Consequently, our goal is settling a stochastic control problem. In the final step, the optimal stopping-time problem will be regarded as a deep learning issue based on the neural network model. Specifically, the neural network model

will act on the policy function Z, which represents the gradient of the solution for the optimal stopping-time problem.

### 2.1. Nonlinear Parabolic Reflected PDEs

　　The PDEs with reflection can be presented by nonlinear parabolic PDEs with minimum constraints. The so-called reflection term forces the solution of the equation to be non-negative, and the reflected part is the minimum power that prevents the solution from leaving the non-negative interval. That is, the reflected PDEs are essentially obstacle problems, and the solutions have only two states: located over the obstacle or on the obstacle. In the beginning, we have some assumptions for the basic parameters and functions which are involved in this problem. Let $T \in (0, \infty), d \in N, f(t, x, y, z)$ is a given non-linear function defined by $f : [0, T] \times R^d \times R \times R^d \to R$, $g(x)$ is a given continuous function defined by $g : R^d \to R$. In this paper, we describe the continuous obstacle by function $h(t, x)$, and the reflected problems for PDEs can be presented by the following scheme:

$$\min\left\{ u(t, x) - h(t, x), -\frac{\partial u}{\partial t} - \mathcal{L}_t u(t, x) - f(t, x, u(t, x), (\nabla u \sigma)(t, x)) \right\} = 0 \qquad (1)$$

where $(t, x) \in (0, T) \times R^d$, $\mathcal{L}_t = \frac{1}{2}\sum_{i,j=1}^{d} \left( \sigma\sigma^*(t, x) \right)_{i,j} \frac{\partial^2}{\partial x_i \partial x_j} + \sum_{i=1}^{d} b_i(t, x) \frac{\partial}{\partial x_i}$.

　　The terminal value of the solution for Equation (1) satisfies $u(T, x) = g(x), x \in R^d$. For Equation (1), there are three further hypotheses as follows:

**Hypothesis 1 (H1).** $b : [0, T] \times R^d \to R^d$, $\sigma : [0, T] \times R^d \to R^{d \times d}$ *are both continuous functions and satisfy:*

$$\|b(t, x) - b(t, y)\| + \|\sigma(t, x) - \sigma(t, y)\| \leq C_1 \|x - y\|;$$

$$\|b(t, x)\|^2 + \|\sigma(t, x)\|^2 \leq C_2^2 \left( 1 + \left\| x^2 \right\| \right).$$

**Hypothesis 2 (H2).** $h : [0, T] \times R^d \to R$ *is a continuous function related to parameters t and x. Meanwhile, it satisfies:*

$$h(t, x) \leq C_3 \left( 1 + |x|^l \right)$$

*where* $h(t, x) \in [0, T] \times R^d, l \in N^+, h(T, x) \leq g(x), x \in R^d$.

**Hypothesis 3 (H3).** $f : [0, T] \times R^d \times R \times R^d \to R$ *is a continuous function, especially,*

$$\left| f(t, x, 0, 0) \right| \leq C \left( 1 + |x|^l \right)$$

*where l is a positive constant.*

$$\left| f(t, x, y, z) - f(t, x, y', z') \right| \leq C_4 \left( |y - y'| + |z - z'| \right)$$

*where* $t \in [0, T], x, z, z' \in R^d, y, y' \in R$.

**Remarks.** *C((·)) > 0 are constants from (H1) to (H3).*

**Proposition 1.** *(Theorem 8.5, 8.6 in* [2]*) It has been proven that the solution u(t,x) for Equation (1) is existing and unique. Furthermore, it is a function with up to polynomial growth.*

### 2.2. From Reflected PDEs to Related Reflected BSDEs

　　In this subsection, we build a connection between the solution of reflected PDEs and reflected BSDEs via the nonlinear reflected Feynman–Kac formula.

Let $(\Omega, \mathcal{F}, \mathbb{P})$ be a given probability space, $\mathcal{B} : [t, T] \times \Omega \to R^d$ be a d-dimensional standard Brownian motion in this space, $\mathcal{F}$ be a normal filtration set generated by $\mathcal{B}$, and continuous functions $b(t, x)$ and $\sigma(t, x)$ satisfy the hypothesis (H1) in Section 2.1. Now, consider a stochastic process with d-dimensional, $\{X_s\} : [t, T] \times \Omega \to R^d$ satisfies:

$$X_t = x + \int_0^t b(r, X_r) dr + \int_0^t \sigma(r, X_r) dB_r \tag{2}$$

From [1], under the assumptions (H1)–(H3), the reflected BSDE (3) has a unique solution represented by a triple $(Y_t, Z_t, K_t)$ [1].

$$\begin{cases} Y_t = g(X_T) + \int_t^T f(r, X_r, Y_r, Z_r) dr + K_T - K_t - \int_t^T (Z_r, dB_r) \\ \int_t^T (Y_S - h(s, X_s)) dK_s = 0, 0 \leq t \leq T \end{cases} \tag{3}$$

Moreover, the solution of Equation (3) satisfies the following three properties:
(C1) $E \int_t^T \left( \lfloor Y_s \rfloor^2 + \lfloor Z_s \rfloor^2 \right) ds < \infty$;
(C2) $Y_t \geq h(t, X_t), 0 < t \leq T$;
(C3) $\{K_t\}$ is continuous and increasing with $K_0 = 0$.

**Proposition 2.** *(Section 8 in* [2]*) Under the assumptions (H1)–(H3), a special relationship between reflected PDEs and reflected BSDEs is established by a classic formula, namely, the Feyman–Kac formula:*

$$u(t, x) = Y_t$$

$$(\nabla u \sigma)(t, x) = Z_t$$

*where* $0 \leq t \leq T, x \in R^d, u(t, x) \in C\left([0, T] \times R^d\right)$.

Consequently, from the above discussions, we transform the reflected problems for PDEs to related reflected problems for BSDEs. That is, $u(t, x)$ can be simulated by $Y_t$. We approximate the reflection term $K_T - K_t$ via the penalization method [2]:

$$K_T - K_t = m \int_t^T (Y_r - h(r, X_r))^- dr \tag{4}$$

where $0 \leq t \leq T$, $m \to +\infty$. Therefore, the reflected BSDE (3) can be approximated by the following scheme:

$$Y_t = g(X_T) + \int_t^T f(Y_r, Z_r) dr + m \int_t^T (Y_r - h(r, X_r))^- dr - \int_t^T (Z_r, dB_r) \tag{5}$$

Obviously, Equation (5) is a general BSDE without any reflection. We use a binary group $(Y_t^m, Z_t^m)$ to illustrate the solution of Equation (5). Then, Equation (1) can be solved approximately by Equation (5). That is, the solution of reflected PDEs can be presented effectively by the solution of related BSDEs. In this work, we simulate the gradient of the solution $u(t, x)$ via a deep neural network rather than solving $\nabla u(t, x)$ directly. The final numerical approximations can be calculated by a series of explicit iteration formats. Details of the whole process will be described in Section 3.

*2.3. Discretizing via Two Approaches*

From Proposition 2, the goal of approximating $u(t, x)$ can be transformed into approximations for $Y_t$. Since the problem we are interested in is the initial value of $u(t, x)$, our goal now is simulating $Y_0$ by the deep learning algorithm based on the deep neural network. To achieve this goal, we discretize Equations (2) and (5) in time dimension by the Euler and C-N schemes, separately. Firstly, we show some general instructions about these two

schemes: the time span is a finite interval from 0 to T, $0 = t_0 < t_1 < \cdots < t_N = T$, with $N \in N^+$.

$$X_{t_{n+1}} - X_{t_n} \approx b(t_n, X_{t_n})\Delta t_n + \sigma(t_n, X_{t_n})\Delta B_n \tag{6}$$

$$K_{t_{n+1}} - K_{t_n} \approx m(Y(t_n, X_{t_n}) - h(t_n, X_{t_n}))^- \Delta t_n \tag{7}$$

$$\begin{aligned} Y(t_{n+1}, X_{t_{n+1}}) \approx {} & Y(t_n, X_{t_n}) - f(t_n, X_{t_n}, Y(t_n, X_{t_n}), (\nabla u\sigma)(t_n, X_{t_n}))\Delta t_n \\ & - m(Y(t_n, X_{t_n}) - h(t_n, X_{t_n}))^- \Delta t_n + \langle (\nabla_x u\sigma)(t_n, X_{t_n}), B_{t_{n+1}} - B_{t_n} \rangle \end{aligned} \tag{8}$$

where $\Delta t_n = t_{n+1} - t_n$, $\Delta B_n = B_{t_{n+1}} - B_{t_n}$

$$\begin{aligned} X_{t_{n+1}} \approx {} & X_{t_n} + \tfrac{1}{2}(b(t_n, X_{t_n}) + b(t_{n+1}, X_{t_{n+1}}))\Delta t_n \\ & + \tfrac{1}{2}(\sigma(t_n, X_{t_n}) + \sigma(t_{n+1}, X_{t_{n+1}}))\Delta B_n \end{aligned} \tag{9}$$

$$K_{t_{n+1}} - K_{t_n} \approx \frac{1}{2}m(Y(t_n, X_{t_n}) + Y(t_{n+1}, X_{t_{n+1}}) - 2h(t_n, X_{t_n}))^- \Delta t_n \tag{10}$$

$$\begin{aligned} Y_{t_{n+1}} \approx {} & Y_{t_n} - \tfrac{1}{2}(f(t_n, X_{t_n}, Y_{t_n}, Z_{t_n}) + f(t_{n+1}, X_{t_{n+1}}, Y_{t_{n+1}}, Z_{t_n}))\Delta t_n \\ & - \tfrac{1}{2}m(Y(t_n, X_{t_n}) + Y(t_{n+1}, X_{t_{n+1}}) - 2h(t_n, X_{t_n}))^- \Delta t_n \\ & + \langle (\nabla_x u\sigma)(t_n, X_{t_n}), B_{t_{n+1}} - B_{t_n} \rangle \end{aligned} \tag{11}$$

Since this discrete format is implicit, i.e., the value at the latter moment cannot be described by a deterministic formula for the value at the previous moment, then next, we perform a second iteration to obtain the explicit format at any time interval $[t_n, t_{n+1}]$, $n = 0, 1, \cdots, N-1$. Let $s = 0, 1, 2, \cdots$, $\varepsilon = 10^{-6}$, for any time interval $[t_n, t_{n+1}]$, we have:

$$X_{t_n}^0 \le X_{t_n}^1 \le \cdots \le X_{t_n}^s \le \cdots \le X_{t_{n+1}}$$

$$K_{t_n}^0 \le K_{t_n}^1 \le \cdots \le K_{t_n}^s \le \cdots \le K_{t_{n+1}}$$

$$Y_{t_n}^0 \le Y_{t_n}^1 \le \cdots \le Y_{t_n}^s \le \cdots \le Y_{t_{n+1}}$$

when $s = 0$:

$$X_{t_{n+1}}^s = X_{t_n}$$

$$Y_{t_{n+1}}^s = Y_{t_n}$$

when $s = 1, 2, \cdots$:

$$\begin{aligned} X_{t_{n+1}}^s \approx {} & X_{t_n} + \tfrac{1}{2}b(t_n, X_{t_n})\left(X_{t_n} + X_{t_{n+1}}^{s-1}\right)\Delta t_n \\ & + \tfrac{1}{2}\sigma(t_n, X_{t_n})\left(X_{t_n} + X_{t_{n+1}}^{s-1}\right)\Delta B_n \end{aligned} \tag{12}$$

$$K_{t_{n+1}}^s - K_{t_n} \approx \frac{1}{2}m\left(Y(t_n, X_{t_n}) + Y\left(t_{n+1}^{s-1}, X_{n+1}^{s-1}\right) - 2h(t_n, X_{t_n})\right)^- \Delta t_n \tag{13}$$

$$\begin{aligned} Y_{t_{n+1}}^s \approx {} & Y_{t_n} - \frac{1}{2}\left(f(t_n, X_{t_n}, Y_{t_n}, Z_{t_n}) + f\left(t_{n+1}^{s-1}, X_{n+1}^{s-1}, Y_{n+1}^{s-1}, Z_{n+1}^{s-1}\right)\right)\Delta t_n \\ & - \frac{1}{2}m\left(Y(t_n, X_{t_n}) + Y\left(t_{n+1}^{s-1}, X_{n+1}^{s-1}\right) - 2h(t_n, X_{t_n})\right)^- \Delta t_n \\ & + \langle (\nabla_x u\sigma)(t_n, X_{t_n}), B_{t_{n+1}} - B_{t_n} \rangle \end{aligned} \tag{14}$$

It should be noted that the condition for stopping the iteration is $\left| Y_{t_{n+1}}^s - Y_{t_{n+1}}^{s-1} \right| < \varepsilon$. As a result, $Y_{t_{n+1}} = Y_{t_{n+1}}^s$

## 3. Numerical Experiments

There are three subsections in this part. Firstly, we introduce a general numerical algorithm for solving high-dimensional nonlinear reflected PDEs, then we illustrate the validity and reliability of this method proposed by us with two examples in Sections 3.2 and 3.3. We operate the equations from continuous time to separation time using the two discrete formats mentioned above, respectively, and the comparison of the significant results will

be described by means of tables. The numerical experiments we present are performed in Python code using TensorFlow on a Lenovo Pro-13 with a Radeon microprocessor.

### 3.1. Deep C-N Algorithm for Solving High-Dimensional Nonlinear Reflected PDEs

In this subsection, we propose a general deep learning framework named the Deep C-N algorithm to calculate high-dimensional nonlinear PDEs with reflection. The issue we are interested in is the initial value of the solutions for Equation (1), i.e., $u(0, x)$. From what we have discussed above, with the intermediary roles of reflected BSDEs, a deterministic relationship is established between the solutions for reflected PDEs and BSDEs. That is: $u(0, x) = Y_{t_0} = Y_{t_0}^m$. In the whole process, the key step is approximating the gradient term of the solution for reflected PDEs, i.e., $(\nabla u \sigma)(t_n, x)$, using multilayer feedback deep neural networks. First of all, we define the definitions of some parameters in this neural networks-based algorithm. Let $\rho \in N$ denote the dimension of reflected PDEs, $\theta \in R^\rho$ is a set of parameters which need to be learned by neural networks, $\left( X_{t_n}^\theta, Y_{t_n}^\theta, Z_{t_n}^\theta \right)$, and $n \in \{0, 1, \cdots, N\}$ is the solution for BSDEs at time $t_n$. For convenience, we use $\left( X_n^\theta, Y_n^\theta, Z_n^\theta \right)$ instead of $\left( X_{t_n}^\theta, Y_{t_n}^\theta, Z_{t_n}^\theta \right)$ in the algorithm description. It should be noted that $X_n^\theta, Y_n^\theta : \{0, 1, \cdots, N\} \times \Omega \to R$, $Z_n^\theta : R^\rho \times \Omega \to R^\rho$. Let $Y^\Theta$ represent $Y_{t_0}^\theta$, which is a part of the initial value of the solution for BSDEs. Then, we have: $Y_{t_0}^\theta \approx Y^\Theta$ and,

$$Y_{n+1}^\theta = Y_n^\theta - f\left(t_n, X_n, Y_n^\theta, Z_n^\theta\right)\Delta t_n - m\left(Y_n^\theta - h(t_n, X_n)\right)^- \Delta t_n + \left\langle Z_n^\theta, \Delta B_n \right\rangle \quad (15)$$

Further, the following relationships hold that:

$$u(t_n, X_{t_n}) = Y_{t_n} \approx Y_n^\theta$$

$$(\nabla u)(t_n, X_{t_n}) = Z_{t_n} \approx Z_n^\theta$$

$$u(0, X_{t_0}) = Y_{t_0} \approx Y^\Theta$$

The loss function is defined by the main square error between the terminal output value calculated via the neural network and the real value at that moment. That is, the loss value is given by

$$\mathbb{E}\left[\left| Y_N^\theta - g(X_N) \right|^2\right]$$

The parameters in neural networks are updating until the loss function is stable. In addition, the existence and uniqueness of the global minimum for Equation (15) is proven by [6]. We choose the stochastic gradient descent (SGD) approach as the optimization algorithm for calculating the loss value. In the back-propagation simulating process, the Adam optimizer is employed to update the parameters layer by layer.

Next, based on the deep neural network technique, the general algorithm framework for solving high-dimensional nonlinear PDEs with reflection is presented (see Figure 1). By dividing the time interval $[0, T]$ by N, a total of $N - 1$ sub-neural networks are computed separately. Assuming that each sub-neural network has H hidden layers, all parameters to be learned will be optimized in each hidden layer. In particular, the results of the operations located in the hidden layer are batch normalized before they are passed through the activation function to the next layer.

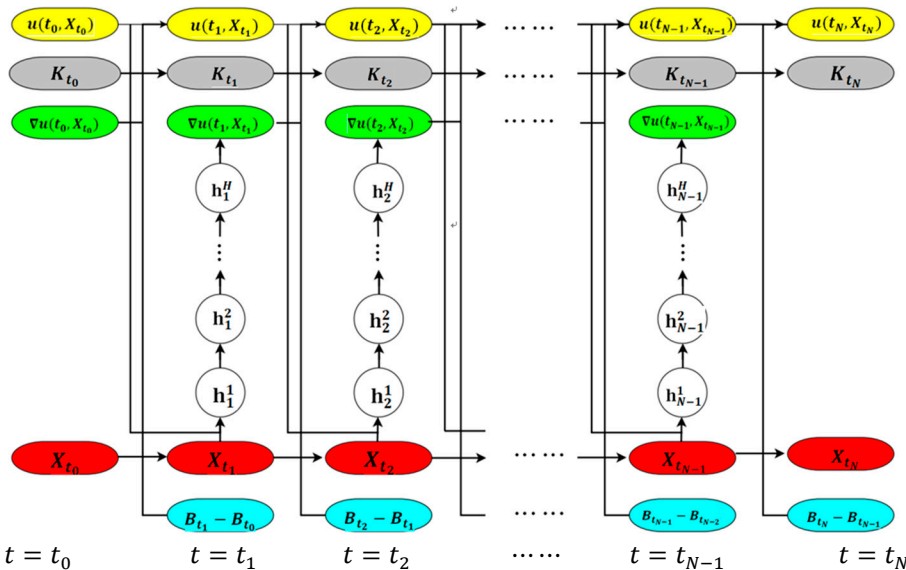

**Figure 1.** Illustration of the neural network framework for solving obstacle problems for PDEs. There are $N-1$ sub-networks in total and H hidden layers in each sub-network. Therefore, there exist $(H+1)(N-1)$ layers in the whole network with parameters that need to be optimized. We divide the time internal $[0, T]$ for intervals and each column for $t = t_1, t_2, \cdots, t_{N-1}$.

The entire deep neural network-based algorithm consists of four types of operations:

1.  $\left(X_{t_n}, B_{t_{n+1}}, B_{t_n}\right) \to X_{t_{n+1}}$ ($n = 1, 2, \cdots N - 1$ and the same settings in 2 to 4) is a forward iterative procedure, which is determined by approximating scheme (6); this procedure does not contain any parameters that need to be optimized.
2.  $\left(K_{t_n}, u(t_n, X_{t_n}), X_{t_n}\right) \to K_{t_{n+1}}$ is a forward iterative procedure too, which is characterized by approximating scheme (7). As in the previous step, no parameters need to be optimized in this operation.
3.  $\left(X_{t_n}, u\left(t_{n-1}, X_{t_{n-1}}\right)\right) \to h_n^1 \to h_n^2 \to (\text{subnetworks}) \cdots \to \nabla u(t_n, X_{t_n})$ is the key step in the whole calculating procedure. Our goal in this step is approximating the spatial gradients, and meanwhile, the weights $\theta_n$ are optimized in the $(N-1)$ sub-networks.
4.  $\left(u(t_n, X_{t_n}), \nabla u(t_n, X_{t_n}), K_{t_{n+1}} - K_{t_n}, B_{t_{n+1}} - B_{t_n}\right) \to u\left(t_{n+1}, X_{t_{n+1}}\right)$ is a forward iteration procedure that yields the neural network's final output as the unique approximation of $u(T, X_T)$, totally characterized by approximating scheme (14).

### 3.2. Allen–Cahn Equation

In this subsection, we consider the solution of PDEs without any boundary. For comparing the results with the Deep BSDE [22] approach, we choose the Allen–Cahn equation as our experimental case. The significant results are presented by figures and tables. The parameters and format of Allen–Cahn equations are as follows:

$\forall t \in [0, T], x, \omega \in \mathbb{R}^d, y \in \mathbb{R}, z \in \mathbb{R}^{1 \times d}$, suppose that $d = 100, N = 20, T = 0.3$, $\mu(t, x) = 0, \sigma(t, x) = \sqrt{2}\omega, X_t = x + \sqrt{2}\omega, f(t, x, y, z) = y - y^3$ is the nonlinear part of the equation, $g(x) = \left[2 + \frac{2}{5}\|x\|_{\mathbb{R}^d}^2\right]^{-1}$ is the value function of the solution at the terminal moment. $u(t, x)$ satisfies:

$$\frac{\partial u}{\partial t}(t, x) + u(t, x) - [u(t, x)]^3 + (\Delta u)(t, x) = 0, \ u(T, x) = g(x) \tag{16}$$

Regarding the settings of the neural network, for comparison with the numerical results in [22], the hyperparameters and the selection of the optimization method are kept the same strategy as that in [22–24], except for the change in the discrete format and the input information of the neural network. The Relu function is chosen as the activation function, $r = 5 \times 10^{-4}$ is the learning rate, and the optimization method uses the Adam

optimizer and batch normalization, with a total of $N-1$ subneural networks for stacking, each containing two hidden layers and each hidden layer contains $d+10$ neurons.

The numerical results of Equation (16) are illustrated by two tables. Specifically, the results calculated by the Deep BSDE method are described in Table 1, and the results calculated by the Deep C-N method are described in Table 2. When solving Equation (16) using the Deep C-N algorithm, we tried appropriate values of iteration steps repeatedly in the interval [2000, 15,000], and found that 10,000 is a suitable number. Too-small values will make the loss value unstable, and too-large values will encounter an overfitting problem. The numerical experiment results shown in the table are based on the mean values of five independent tests. The true value of Equation (16) is 0.052802, which is derived from the Branching diffusion algorithm in [22].

**Table 1.** The numerical results of Deep BSDE algorithm.

| Number of Iteration Step | Mean of $Y_0$ | Standard Deviation of $Y_0$ | Relative $L^1$-Approximate Error | Relative $L^1$-Approximate Error Associated with $Y_0$ | Mean Value of Loss Function | Standard Deviation of Loss Function |
|---|---|---|---|---|---|---|
| 0 | 0.4740 | 0.0514 | 7.9775 | 0.9734 | 0.11630 | 0.02953 |
| 1000 | 0.1446 | 0.0340 | 1.7384 | 0.6436 | 0.00550 | 0.00344 |
| 2000 | 0.0598 | 0.0058 | 0.1318 | 0.1103 | 0.00029 | 0.00006 |
| 3000 | 0.0530 | 0.0002 | 0.0050 | 0.0041 | 0.00023 | 0.00001 |
| 4000 | 0.0528 | 0.0002 | 0.0030 | 0.0022 | 0.00020 | 0.00001 |

**Table 2.** The numerical results of Deep C-N algorithm.

| Number of Iteration Step | Mean of $Y_0$ | Standard Deviation of $Y_0$ | Relative $L^1$-Approximate Error | Relative $L^1$-Approximate Error Associated with $Y_0$ | Mean Value of Loss Function | Standard Deviation of Loss Function |
|---|---|---|---|---|---|---|
| 0 | 0.5021 | 0.0791 | 0.2979 | 0.449313 | 0.137191 | 0.043493 |
| 2000 | 0.0659 | 0.0083 | 0.0131 | 0.011521 | 0.000407 | 0.000142 |
| 4000 | 0.0569 | 0.0021 | 0.0002 | 0.000040 | 0.000201 | 0.000027 |
| 6000 | 0.0531 | 0.0002 | 0.0002 | 0.000013 | 0.000118 | 0.000240 |
| 8000 | 0.0529 | 0.0002 | 0.0002 | 0.000156 | 0.000055 | 0.000012 |
| 10,000 | 0.0528 | 0.0001 | 0.0001 | 0.000117 | 0.000030 | 0.000010 |

The two tables above illustrate that the numerical results of the Deep C-N algorithm are better than the Deep BSDE algorithm. The most significant aspect is that the computational accuracy of the Deep C-N algorithm is better than that of the Deep BSDE algorithm. This is demonstrated by the fact that the loss value of the Deep BSDE algorithm is approaching $2 \times 10^{-4}$, while the value of the Deep C-N algorithm is approaching $3 \times 10^{-5}$, which means that the accuracy is improved by nearly an order of magnitude. In other aspects, the standard deviation of $Y_0$ of the Deep C-N method is smaller than the Deep BSDE method, which demonstrates that the approximating stability of the Deep C-N method is superior to the Deep BSDE method. In addition, the smaller relative $L^1$-approximate error, to a certain extent, reflects the strengths of the model.

We now discuss the stability and accuracy of these two numerical models via eight equations with 50 to 400 dimensions. In Figures 2 and 3, it can be clearly seen that the numerical results of the Deep C-N algorithm are more compact. In particular, the loss function curves of the Deep C-N algorithm are close to overlapping when the dimensionality is greater than 100, while the Deep BSDE algorithm shows a larger difference. This indicates that the Deep C-N model has better stability than the Deep BSDE model when dealing with high-dimensional problems. Regarding the approximation accuracy of the model, Table 3 shows that in tests for different dimensions, the loss values of the Deep C-N model are smaller than the corresponding results of the Deep BSDE model. This indicates that the computational accuracy of the Deep C-N algorithm is higher in the interval of 50 to 400 dimensions.

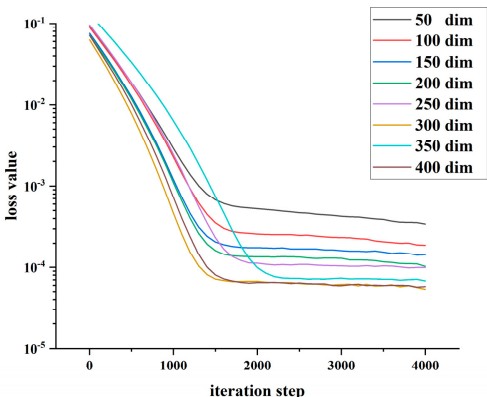

**Figure 2.** Loss value of different dimensions for Deep BSDE algorithm.

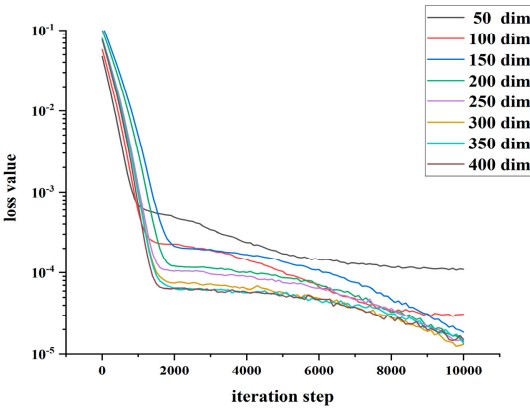

**Figure 3.** Loss value of different dimensions for Deep C-N algorithm.

**Table 3.** Convergence levels of loss function of the two methods in different dimensions.

| Dimensions | 50 | 100 | 150 | 200 | 250 | 300 | 350 | 400 |
|---|---|---|---|---|---|---|---|---|
| Deep BSDE | $3.415 \times 10^{-4}$ | $1.886 \times 10^{-4}$ | $1.44 \times 10^{-4}$ | $1.029 \times 10^{-4}$ | $9.973 \times 10^{-5}$ | $5.330 \times 10^{-5}$ | $7.789 \times 10^{-5}$ | $5.774 \times 10^{-5}$ |
| Deep C-N | $1.095 \times 10^{-4}$ | $3.095 \times 10^{-5}$ | $1.853 \times 10^{-5}$ | $1.435 \times 10^{-5}$ | $1.398 \times 10^{-5}$ | $1.303 \times 10^{-5}$ | $1.337 \times 10^{-5}$ | $1.506 \times 10^{-5}$ |

### 3.3. American Options

In this subsection, we consider the solution of reflected PDEs with continuous boundary. We select the case of pricing American options in order to make a comparison with the test results in [4]. The significant results are presented as well. The parameters and format of the American options are as follows:

$\forall t \in [0, T], x, \omega \in R^d, y \in R, z \in R^{1 \times d}$, to be consistent with [4], suppose that d = 5, 10,20,40 , N = 20, T = 1, K = 1, r = 0.05, is the risk-free interest rate, $\sigma(t, x) = 0.2$, $dX_t = \mu(t, x)X_t dt + \sigma(t, x)X_t dW_t$, where $W_t$ is a d-dimension Brownian motion with d dimensions, $g(x) = \left(K - \prod_{i=1}^{d} X_t^i\right)_+$ is the payoff of option. At time t, the value of an American option $u(t, x)$ satisfies:

$$u(t, x) = sup_{\tau \in \mathcal{T}_{t,T}} E\left[e^{-r\tau} g(X_\tau)\right]$$

where $\mathcal{T}_{t,T}$ is defined as the set of stopping times; it is also a solution to the PDEs following with a boundary

$$\begin{cases} min[-\partial_t u - \mathcal{L}u, u - g] = 0, & on [0, T) \times (0, \infty)^d \\ u(T, .) = g, & on (0, \infty)^d \end{cases} \quad (17)$$

where $\mathcal{L}u(t,x) = \frac{1}{2}\sum_{i=1}^{d} \sigma_i^2 x_i^2 D_{x_i}^2 u(t,x) + r\sum_{i=1}^{d} x_i D_{x_i} u(t,x) - ru(t,x)$

Now we display the numerical results for pricing American options by three models:

The results in Table 4 show that both the Deep C-N method and the RDBDP method have great approximation accuracy up to 40 dimensions, while the Deep BSDE method has exploding loss values or eventually leads to the wrong results. Therefore, it is concluded that both the Deep C-N method and the RDBDP method have excellent performance when dealing with reflected PDE problems up to 40 dimensions, while the Deep BSDE method cannot solve the high-dimensional reflected PDE problems.

**Table 4.** Numerical solutions for 3 models.

| Models | Dimensions | Value | Reference | Relative Error |
|---|---|---|---|---|
| Deep C-N | 5 | 0.10720 | 0.10738 | 0.17% |
| RDBDP | 5 | 0.10657 | 0.10738 | 0.75% |
| Deep BSDE | 5 | NC | 0.10738 | NC |
| Deep C-N | 10 | 0.12687 | 0.12996 | 2.38% |
| RDBDP | 10 | 0.12829 | 0.12996 | 1.29% |
| Deep BSDE | 10 | NC | 0.12996 | NC |
| Deep C-N | 20 | 0.15140 | 0.15100 | 0.27% |
| RDBDP | 20 | 0.14430 | 0.15100 | 4.38% |
| Deep BSDE | 20 | NC | 0.15100 | NC |
| Deep C-N | 40 | 0.16213 | 0.16800 | 3.49% |
| RDBDP | 40 | 0.16167 | 0.16800 | 3.77% |
| Deep BSDE | 40 | NC | 0.16800 | NC |

It has been proved in [4] that the RDBDP algorithm has an obvious limitation when solving PDEs with reflection over 40 dimensions. Next, we demonstrate through Figure 4 that the Deep C-N method still has great stability when dealing with reflected PDE problems with more than 40 dimensions.

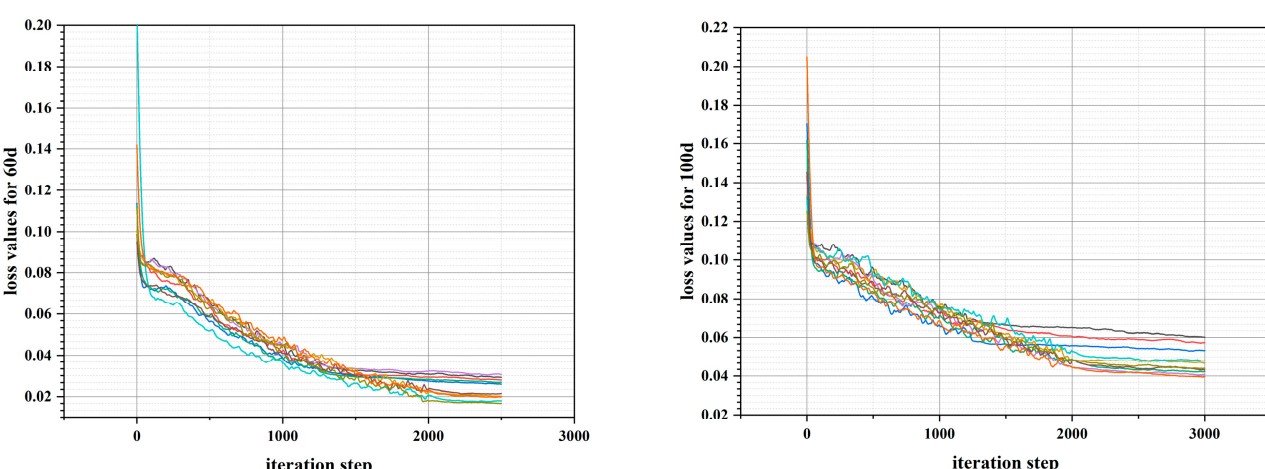

**Figure 4.** Loss values for 10 times based on Deep C-N algorithm.

## 4. Discussion and Conclusions

In this paper, we proposed a deep learning-based numerical algorithm (named the Deep C-N method) for solving high-dimensional nonlinear reflected PDEs. Through numerical experiments, we found that the Deep C-N method has great approximating accuracy and computational stability in solving reflected PDEs of up to 100 dimensions. To obtain this result, we created a hybrid discrete format to solve some specific reflection problems. Compared with a single format, we found that this hybrid discretization strategy effectively overcomes the problem of loss function explosion or computational paralysis during the neural network computation. In addition, we used the penalization method

to approximate the reflection terms instead of simulating them directly with numerical algorithms. The results show that the relative errors of the Deep C-N method are all below 4% when solving up to 40-dimensional reflected PDEs. As the dimension of the equation increases, the value of the neural network loss function shows an increasing trend, and the value of the neural network loss function stabilizes between 0.04 and 0.06 when the dimension rises to 100 dimensions. The Deep C-N method encountered challenges when trying to solve the reflection PDE above 100 dimensions. The first challenge is the computation time rising significantly with the increase in dimensions. The second is the computational accuracy decreasing as the dimensions increase. We support that these two problems are the most common difficulties encountered when dealing with high-dimensional issues, and we can say that the key to developing numerical algorithms for high-dimensional problems is to overcome these two problems.

The method proposed in this paper contributes in two ways. First, in solving high-dimensional reflection PDEs, one of the successful deep learning algorithms is the RDBDP algorithm proposed by Huré et al. [4], which can solve up to 40-dimensional reflected PDEs. The method proposed in this paper successfully extends the dimensions of the equation up to 100 dimensions. Second, in solving high-dimensional PDEs without reflection terms, the method proposed in this paper possesses higher approximating accuracy compared with the Deep BSDE methods proposed by Han et al. [22], Beck et al. [23], and Han et al. [24]. Therefore, the Deep C-N method proposed in this paper can be used to solve both reflected PDEs of up to 100 dimensions and high-dimensional nonlinear PDEs (tested up to 400 dimensions).

Since the discrete strategy we use has more steps, this inevitably leads to an increase in training difficulty and computational cost. One of the most direct evidence is that when approximating the 100-dimensional Allen–Cahn equation, the results can be converged in 426 s with 4000 training sessions using the Deep BSDE method, while with the Deep C-N method proposed in this paper, 10,000 training sessions are required to make the results converge in 3508 s (the exact running time varies with the machine).

**Author Contributions:** Conceptualization, J.Y.; methodology, J.Y. and X.S.; software, X.S. and X.Z.; validation, X.S. and J.Y.; formal analysis, X.S. and J.Y.; investigation, X.S. and X.Z.; resources, J.Y. and R.T.; data curation, X.S. and X.Z.; writing—original draft preparation, X.S.; writing—review and editing, J.Y.; visualization, X.S. and X.Z.; supervision, R.T.; project administration, J.Y.; funding acquisition, J.Y. All authors have read and agreed to the published version of the manuscript.

**Funding:** This research received no external funding.

**Data Availability Statement:** No new data were created or analyzed in this study. Data sharing is not applicable to this article.

**Conflicts of Interest:** The authors declare no conflict of interest.

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
