# Peer review of "Solve High-Dimensional Reflected Partial Differential Equations by Neural Network Method"

_mca, doi:10.3390/mca28040079_

Round 1
Reviewer 1 Report
The authors solved high-dimensional reflected partial differential equations by neural network method. By numerical experiments, the authors found that the Deep C-N method has great approximating accuracy and computational. Additionally, the authors used penalization method to approximate the reflection terms instead of simulating them directly with numerical algorithms. The results showed that the relative errors of the Deep C-N method are all below 4% when solving up to 40-dimensional reflected PDEs. The manuscript is well-written and I am in favour of publication of the manuscript.
The quality of English is good.
Reviewer 2 Report
This paper proposed the Deep C-N method for solving the obstacle problems for high-dimensional non-linear parabolic PDE. The algorithm is well-defined. I suggest accepting it after minor revision. My comments are listed as follows:
1. For the relation between PDE (1) and BSDE (3), could you explain more on the part of K? What is the intuition of K in the PDE? Why can it be approximated using (4)?
2. You mentioned two numerical schemes for approximating BSDEs: Euler and CN. You used CN and reported more accurate results compared with the literature. Does the replacement of the numerical scheme cause this extra accuracy? Can we infer the results will be better using a more accurate numerical scheme?
3. The implicit scheme CN causes an inner iteration. Also, more steps in the scheme cause more dependence of the loss function on the neural network, which may increase the training difficulty, in terms of back-propagation computational cost and the potential of vanishing gradient. Could you add some numerical results or discussion about this part? How does your method compare with the literature in terms of computational cost?
4. In terms of literature, there are other neural network methods for solving HJ PDEs. Please check the papers of Stan Osher, Jerome Darbon, Peter Dower, Wei Kang, and others.
5. Can you add more details on the numerical parts? For instance, when you compare with other NN methods, do they have similar sizes (how many layers and number of parameters)? Also, what is Figure 5? Is it the result of the last experiment (could you also add a legend in those figures)?
6. When you train the NN, do you sample the initial position of Xt? If not, could you add some explanation to this part? If you do the sampling for X0, does this sampling distribution influence the results a lot?
7. When you set the stopping criteria for the inner iteration of the CN scheme, you check the difference of Yt. Is it possible that Yt has small errors but other processes (Xt and Kt) may have big errors?
8. The loss function only depends on the terminal condition. When we did experiments on NN solving PDEs, we found that other terms in loss function may help to stabilize the training process. Have you tried other terms in loss?
Round 2
Reviewer 2 Report
The authors have replied all the questions in the last round and the answers look clear to me. I would suggest accepting the manuscript.